# Post Hoc Analysis of a Randomized Controlled Trial on Fasting and Plant-Based Diet in Rheumatoid Arthritis (NutriFast): Nutritional Supply and Impact on Dietary Behavior

**DOI:** 10.3390/nu15040851

**Published:** 2023-02-07

**Authors:** Anika M. Hartmann, Marina D’Urso, Melanie Dell’Oro, Daniela A. Koppold, Nico Steckhan, Andreas Michalsen, Farid I. Kandil, Christian S. Kessler

**Affiliations:** 1Department of Dermatology, Venereology and Allergology, Charité—Universitätsmedizin Berlin, Corporate Member of Freie Universität Berlin and Humboldt-Universität zu Berlin, 10117 Berlin, Germany; 2Institute for Social Medicine, Epidemiology and Health Economics, Charité—Universitätsmedizin Berlin, Corporate Member of Freie Universität Berlin and Humboldt-Universität zu Berlin, 10117 Berlin, Germany; 3Department of Gastroenterology, Infectiology and Rheumatology, Charité—Universitätsmedizin Berlin, Corporate Member of Freie Universität Berlin and Humboldt-Universität zu Berlin, 10117 Berlin, Germany; 4Department of Internal and Integrative Medicine, Immanuel Hospital Berlin, 10117 Berlin, Germany; 5Connected Healthcare, Hasso Plattner Institute, University of Potsdam, 10117 Potsdam, Germany; 6Department of Paediatric Oncology/Haematology, Otto-Heubner Centre for Paediatric and Adolescent Medicine (OHC), Charité—Universitätsmedizin Berlin, Corporate Member of Freie Universität Berlin and Humboldt-Universität zu Berlin, 10117 Berlin, Germany

**Keywords:** intermittent fasting, caloric restriction, plant-based diet, anti-inflammatory diet, time-restricted eating, macronutrients, micronutrients, cluster analysis, food record

## Abstract

This study aimed at comparing the nutrient supply and dietary behaviors during a plant-based diet (PBD) combined with time-restricted eating (TRE) to standard dietary recommendations in rheumatoid arthritis patients. In this open-label, randomized, controlled clinical trial, patients were assigned to either a 7-day fast followed by an 11-week PBD including TRE (A) or a 12-week anti-inflammatory diet following official German guidelines (German Nutrition Society, DGE) (B). Dietary habits were assessed by 3-day food records at weeks -1, 4 and 9 and food frequency questionnaires. 41 out of 53 participants were included in a post-hoc per protocol analysis. Both groups had similar energy, carbohydrate, sugar, fiber and protein intake at week 4. Group A consumed significantly less total saturated fat than group B (15.9 ± 7.7 vs. 23.2 ± 10.3 g/day; *p* = 0.02). Regarding micronutrients, group B consumed more vitamin A, B_12_, D, riboflavin and calcium (each *p* ≤ 0.02). Zinc and calcium were below recommended intakes in both groups. Cluster analysis did not show clear group allocation after three months. Hence, dietary counselling for a PBD combined with TRE compared to a standard anti-inflammatory diet does not seem to lead to two different dietary clusters, i.e., actual different dietary behaviors as expected. Larger confirmatory studies are warranted to further define dietary recommendations for RA.

## 1. Introduction

Rheumatoid arthritis (RA) is an inflammatory joint disease whose etiology is still largely unknown. Accordingly, there is no causal therapy yet [1]. Since RA is not considered curable, treatment goals focus on achieving clinical remission and preventing organ damage [2]. In recent years, numerous new disease-modifying antirheumatic drugs (DMARDs) have been developed in the form of targeted antibody therapies that can block various inflammatory mechanisms. Although better treatment outcomes have been achieved compared with conventional therapies, patients respond differently to these new therapies. Therefore, the optimal drug for each patient must be determined through individualized treatment attempts.

In search for complementary therapeutic options, many studies have been conducted on dietary influences in patients with RA [3]. Red meat, alcohol, coffee, sugars and sweets were the foods most often reported to worsen RA symptoms [4], while foods rich in vitamins, minerals and omega-3 fatty acids seem to lead to clinical improvements in RA symptoms [5,6]. Although most patients with RA benefit from a short fasting period, almost all relapse when reintroducing usual diets [7,8]. A reduction in disease activity in patients with RA can be accomplished by fasting followed by an individually adjusted vegetarian diet for up to one year [9].

We have previously reported on the benefits of fasting followed by a plant-based diet (PBD) on disease activity and cardiovascular (CV) risk factors in RA. The positive impact was comparable to that of an anti-inflammatory diet based on the recommendations of the German Nutrition Society (Deutsche Gesellschaft für Ernährung, DGE) [10]. The similar performance of the two groups prompted us to question (i) whether the nutritional counselling provided was reflected in actual dietary behaviors and (ii) whether it actually resulted in the two distinctively different behavioral groups that were intended by randomization.

Additionally, participants assigned to the fasting and PBD group in this trial experienced greater weight loss than the DGE group within the normal BMI range. However, it remains uncertain whether, in this case, weight reduction is attributable to reduced daily energy intake, altered diet composition or medium-term sustained fasting effects. Prolonged fasting is known for its blood-pressure-lowering and weight-reducing effects via increased water excretion [11,12,13,14,15]. However, this is usually reversible within 12 weeks. Additionally, time-restricted eating may also reduce body weight [16].

Our objectives were (i) to investigate whether the delivered dietary intervention corresponded to the actual dietary behavior and (ii) to compare the influences of the two different dietary approaches on nutrient supply.

## 2. Materials and Methods

### 2.1. Study Design and Population

The present study is a post-hoc analysis of a randomized controlled clinical trial (NutriFast) that we reported earlier [10]. The study design has been described elsewhere in detail [17]. In brief, we performed a monocentric randomized controlled exploratory trial with a parallel group design at the Immanuel Hospital Berlin, where the Outpatient Department for Integrative Medicine of the Institute of Social Medicine, Epidemiology and Health Economics of Charité—Universitätsmedizin Berlin is seated. The study protocol was approved by the institutional review board of Charité Universitätsmedizin Berlin (Charitéplatz 1, 10117 Berlin, ID: EA4/005/17), was registered with ClinicalTrials.gov (https://clinicaltrials.gov/; accessed on 3 February 2023; ClinicalTrials ID: NCT03856190) and was carried out according to the standards of the Declaration of Helsinki.

Between March 2019 and November 2020, we enrolled and randomized participants with RA on stable medication who were healthy enough to participate in an outpatient fasting program. Key exclusion criteria included previous eating disorders, underweight status, a strict plant-based diet and/or fasting within the last six months, as well as changes in therapy with DMARDs in the last 8 weeks before enrolment [18].

### 2.2. Dietary Intervention

Participants were assigned to one of two treatment arms in a 1:1 ratio, which have been extensively described elsewhere [18]. In summary, the interventional study arm consisted of a 7-day fast followed by a whole-food plant-based diet (PBD) for 11 weeks. The PBD integrated the concept of time-restricted eating (TRE, 16/8 h). Additionally, it was enriched with spices and herbs known for their anti-inflammatory potential, such as cinnamon, turmeric and cilantro [19,20,21]. The second study arm adopted a guideline-based diet according to the DGE for 12 weeks. Essentially, this regimen recommends a reduced intake of arachidonic acid, which is mainly contained in foods of animal origin, in order to achieve anti-inflammatory effects.

We delivered the nutrition interventions as an intensive, group-based behavioral intervention in an outpatient setting. Participants received 3 individual and 9 group dietary counselling sessions over a period of 3 months.

### 2.3. Study Variables and Instruments

We assessed dietary intake in the NutriFast study using a food frequency questionnaire (FFQ) for the cluster analysis and 3-day food records for nutrient analysis. The FFQ queries the consumption frequency of certain spices and herbs at baseline and at weeks 6 and 12. Because of the scheduled counselling sessions based on the food records, these were collected at different times to the FFQ. Three 3-day food records were assessed for each patient at study week -1 (baseline) and at study weeks 4 and 9. Participants weighed and recorded all foods and beverages consumed using an electronic kitchen scale on three consecutive days (2 weekdays and one weekend day). When exact weighing was not possible—e.g., in the case of eating out—household measures (e.g., spoons, cups) and photos of the dish allowed semi-quantitative recording. A trained research team member coded food records according to a standardized data entry manual using the NutriGuide^®^ software (Version 4.8, Plus, NutriScience, Stuttgart, Germany). The composition of self-prepared dishes was calculated from the raw ingredients of the reported recipes/dishes.

Absolute dietary intakes were compared with the current dietary reference intake (DRI) published by the Societies for Nutrition in Germany (DGE), Austria (Österreichische Gesellschaft für Ernährung, ÖGE) and Switzerland (Schweizerische Gesellschaft für Ernährung, SGE), being the so-called D-A-CH (D—Deutschland, Germany, A—Austria, CH—Confoederatio Helvetica, Switzerland) recommendations [22].

### 2.4. Statistical Analysis

Total energy (kcal/day) and nutrient intakes were calculated for each patient using the software NutriGuide^®^ (Version 4.8, Plus, Nutri-Science, Stuttgart, Germany), which works with the database of the German Federal Food Code (BLS) of the German Federal Institute of the Protection of Consumer Health and Veterinary Medicine (BGVV). The daily recommended intake was adjusted to gender, age and physical activity level (PAL) of the participant. The PAL refers to the daily additional energy consumption to be expended for physical activities in relation to resting energy consumption. A PAL of, e.g., 1.6 indicates a sedentary lifestyle with occasionally additional energy expenditure for walking and standing activity, as in the case of laboratory assistants, students and assembly line workers [22]. The displayed reference values for nutrient intake are those for healthy adult women with a PAL of 1.6 and were taken from the DGE. For the statistical analysis of the continuous variables, the Mann–Whitney U-Test was used considering that the variables may not have a normal distribution. Significant results (*p* ≤ 0.05) have been bolded. All statistical analyses were performed using SPSS^®^ Statistics (Version 27.0 Armonk, NY, USA: IBM Corp.).

### 2.5. Cluster Analysis

We performed a posteriori cluster analysis to examine the impact of the initially constructed dietary interventions on eating habits over the course of the study [23]. Hierarchical exploratory cluster analyses were conducted for two timepoints: the changes in diet between baseline and week 6, as well as between baseline and week 12, across the 33 weighed and recorded foods and beverages consumed on three consecutive days around baseline, week 6 and week 12. Affiliation with one of the two clusters (at the topmost hierarchical step) was then compared to the affiliation of the two original intervention groups via the X² test.

## 3. Results

Of 53 randomized participants, 50 participants completed the dietary interventions (25 per group) (Figure 1). Of these, 41 participants had complete longitudinal data sets in the nutritional per protocol analysis. All analyzed study participants were female, of which 24 participants were assigned to the fasting and PBD group and 17 participants to the DGE group. We performed the nutritional analysis based on the 3-day food records at study weeks -1, 4 and 9. The results of the statistical analyses of each group were compared for each visit with the DGE reference values for healthy adult women (Table 1, Table 2 and Table 3) [22].

### 3.1. Nutrient Supply

There were no significant differences regarding nutrient supply between the DGE group and the fasting and PBD group at study week -1, except for riboflavin intake (1.1 ± 0.5 mg/day vs. 0.8 ± 0.6 mg/day, *p* = 0.02, respectively). The total energy and nutrient parameters at study week -1 are displayed in Table 1.

At study weeks 4 and 9, the intake of various analyzed macro- and micronutrients differed between the two groups (see Table 2 and Table 3, and Figure 2).

**Table 2 nutrients-15-00851-t002:** Nutrient supply of participants with RA at study week 4 as well as the DGE reference values for adult women [22].

Nutrients	Overall Mean ± SD (*n* = 41)	Fasting + PBD Mean ± SD (*n* = 24)	DGE Mean ± SD (*n* = 17)	Reference Values	*p*-Value
Energy intake, kcal/day	1909.2 ± 744.3	1762.7 ± 690.8	2115.9 ± 788.4	2100.0	0.17
Carbohydrates, g/day	222.2 ± 132.4	197.8 ± 93.9	256.8 ± 170.3	256.1	0.30
Sugar, g/day	72.6 ± 54.8	75.1 ± 64.4	68.1 ± 32.3	61.0	0.89
Total fat, g/day	77.1 ± 29.3	76.6 ± 33.8	77.8 ± 22.4	69.2	0.43
Saturated	20.4 ± 9.5	17.3 ± 8.5	24.7 ± 9.5	16.2–23.1	**0.02**
Monounsaturated	20.0 ± 13.0	21.5 ± 15.5	17.5 ± 5.8	34.6–46.2	0.94
Polyunsaturated	17.6 ± 9.3	19.0 ± 9.4	15.6 ± 9.1	18.5–27.7	0.23
Protein, g/day	59.4 ± 24.6	50.9 ± 22.4	71.5 ± 23.0	48.0	**0.003**
Dietary fiber, g/day	31.5 ± 13.6	31.3 ± 13.9	31.7 ± 13.4	35.1	0.77
Vitamin E, mg/day	13.1 ± 7.0	14.3 ± 7.8	11.4 ± 5.4	14.0	0.27
Vitamin C, mg/day	129.8 ± 75.8	130.5 ± 88.6	128.8 ± 55.6	95.0	0.58
Vitamin D, µg/day	3.2 ± 4.4	1.7 ± 1.4	5.9 ± 6.6	2.0–4.0	**0.04**
Thiamine, mg/day	0.8 ± 0.4	0.9 ± 0.4	0.8 ± 0.4	1.0	0.69
Riboflavin, mg/day	0.9 ± 0.5	0.8 ± 0.4	1.0 ± 0.6	1.1	0.18
Vitamin B6, mg/day	1.1 ± 0.5	1.0 ± 0.4	1.2 ± 0.6	1.4	0.49
Biotin, µg/day	23.6 ± 19.6	23.6 ± 21.2	23.5 ± 17.3	40.0	0.84
Folate, µg/day	214.0 ± 104.8	213.3 ± 90.7	215.4 ± 131.3	300.0	0.77
Vitamin B12, µg/day	1.5 ± 2.2	0.6 ± 0.8	3.0 ± 3.0	4.0	**0.001**
Vitamin A, mg/day	0.1 ± 0.1	0.1 ± 0.1	0.2 ± 0.2	0.7	**0.02**
Vitamin K, µg/day	68.3 ± 52.2	62.5 ± 40.3	79.0 ± 69.8	60.0	0.83
Zinc, mg/day	6.5 ± 4.2	6.5 ± 4.8	6.4 ± 3.1	8.0	0.69
Magnesium, mg/day	336.5 ± 194.0	354.3 ± 222.9	303.9 ± 127.5	300.0	0.80
Copper, µg/day	1630.5 ± 693.3	1631.9 ± 721.9	1628.0 ± 668.6	1000–1500	0.89
Calcium, mg/day	580.1 ± 336.4	513.1 ± 319.8	702.9 ± 344.6	1000.0	**0.04**

Values reported are means ± SD or *n* (%) unless specified otherwise. Reference values are displayed for orientation only. Significant values (*p* ≤ 0.05) of inter-group comparisons are bolded. DGE, German Nutrition Society; kcal, kilocalorie; PBD, plant-based diet.

**Table 3 nutrients-15-00851-t003:** Nutrient supply of participants with RA at study week 9 as well as the DGE reference values for adult women [22].

Nutrients	Overall Mean ± SD (*n* = 41)	Fasting + PBD Mean ± SD (*n* = 24)	DGE Mean ± SD (*n* = 17)	Reference Values	*p*-Value
Energy intake, kcal/day	1810.2 ± 585.0	1764.4 ± 492.0	1874.8 ± 707.1	2100.0	0.96
Carbohydrates, g/day	211.0 ± 99.6	202.4 ± 78.0	223.2 ± 125.6	256.1	0.79
Sugar, g/day	82.6 ± 52.0	85.6 ± 61.3	76.7 ± 25.9	61.0	0.71
Total fat, g/day	72.0 ± 24.9	73.3 ± 25.2	70.3 ± 25.0	69.2	0.94
Saturated	18.9 ± 9.5	15.9 ± 7.7	23.2 ± 10.3	16.2–23.1	**0.02**
Monounsaturated	19.4 ± 10.2	21.4 ± 11.8	15.5 ± 4.2	34.6–46.2	0.38
Polyunsaturated	17.0 ± 9.5	17.6 ± 9.4	16.1 ± 9.7	18.5–27.7	0.61
Protein, g/day	56.3 ± 19.8	52.7 ± 17.9	61.3 ± 21.7	48.0	0.34
Dietary fiber, g/day	31.5 ± 18.5	29.1 ± 10.2	35.0 ± 26.2	35.1	0.96
Vitamin E, mg/day	12.5 ± 6.8	13.8 ± 6.7	10.7 ± 6.6	14.0	0.05
Vitamin C, mg/day	159.1 ± 124.6	145.8 ± 96.8	117.8 ± 157.1	95.0	0.69
Vitamin D, µg/day	1.7 ± 2.0	1.1 ± 0.8	2.9 ± 3.1	2.0–4.0	**0.01**
Thiamine, mg/day	0.8 ± 0.3	0.8 ± 0.3	0.8 ± 0.3	1.0	0.55
Riboflavin, mg/day	0.9 ± 0.3	0.8 ± 0.3	1.0 ± 0.4	1.1	**0.02**
Vitamin B6, mg/day	1.1 ± 0.5	1.1 ± 0.4	1.2 ± 0.6	1.4	0.81
Biotin, µg/day	29.8 ± 34.7	32.5 ± 41.7	24.2 ± 12.2	40.0	0.69
Folate, µg/day	226.8 ± 143.9	231.2 ± 133.2	218.1 ± 169.1	300.0	0.16
Vitamin B12, µg/day	1.0 ± 1.3	0.4 ± 0.6	2.3 ± 1.5	4.0	**0.00**
Vitamin A, mg/day	0.1 ± 0.1	0.1 ± 0.1	0.2 ± 0.1	0.7	**0.01**
Vitamin K, µg/day	98.8 ± 109.1	108.9 ± 128.4	78.5 ± 52.0	60.0	0.84
Zinc, mg/day	5.9 ± 2.2	5.6 ± 1.8	6.4 ± 2.8	8.0	0.70
Magnesium, mg/day	320.6 ± 118.2	319.6 ± 95.5	322.7 ± 159.4	300.0	0.46
Copper, µg/day	1611.1 ± 531.8	1646.8 ± 434.6	1539.8 ± 704.6	1000–1500	0.24
Calcium, mg/day	569.8 ± 278.5	479.9 ± 187.0	731.6 ± 357.3	1000.0	**0.02**

Values reported are means ± SD or *n* (%) unless specified otherwise. Reference values are displayed for orientation only. Significant values (*p* ≤ 0.05) of inter-group comparisons are bolded. DGE, German Nutrition Society; kcal, kilocalorie; PBD, plant-based diet.

First, the DGE group had a higher intake of saturated fatty acids, above the reference range (24.7 ± 9.5 g/day vs. 17.3 ± 8.5 g/day, *p* = 0.02, respectively). However, we did not observe this difference in total daily fat consumption.

The DGE group consumed more daily protein than the fasting and PBD group, even above the daily recommended intake, at week 4 (71.5 ± 23.0 g/day vs. 50.9 ± 22.4 g/day, *p* = 0.003, respectively) through to week 9 (61.3 ± 21.7 g/day vs. 52.7 ± 17.9 g/day, *p* = 0.34, respectively).

Furthermore, the DGE group ingested more vitamin D than the fasting and PBD group and even exceeded the recommended daily intake group at week 4 (5.9 ± 6.6 μg/day vs. 1.7 ± 1.4 μg/day, *p* = 0.04, respectively). At week 9, dietary vitamin D uptake was within the reference range in the DGE group but below it in the fasting and PBD group (2.9 ± 3.1 μg/day vs. 1.1 ± 0.8 μg/day, *p* = 0.01, respectively).

Both groups did not reach the recommended daily intake of vitamin B12. However, the DGE group yielded a more sufficient intake than the fasting and PBD group at both timepoints (3.0 ± 3.0 µg/day vs. 0.6 ± 0.8 µg/day, *p* = 0.001, respectively, at week 4 and 2.3 ± 2.5 µg/day vs. 0.4 ± 0.6 µg/day, *p* = 0.0001, respectively, at week 9).

Neither group reached the recommended daily allowance of vitamin A, although the DGE group achieved higher amounts than the fasting and PBD group (0.2 ± 0.2 mg/day vs. 0.1 ± 0.1 mg/day, *p* = 0.02, respectively, at week 4 and 0.2 ± 0.1 mg/day vs. 0.1 ± 0.1 mg/day, *p* = 0.01, respectively, at week 9).

As for minerals, only calcium intake stood out, being below the recommended reference values in both groups, even though it was higher in the DGE group by week 9 (513.1 ± 319.8 mg/day in the fasting and PDB group vs. 702.9 ± 344.6 mg/day in the DGE group, *p* = 0.04).

### 3.2. Cluster Analysis

The study population was divided into individual, non-overlapping clusters based on food consumption. This resulted in every study participant being allocated to a specific cluster.

However, hierarchical exploratory cluster analyses did not reveal any significant difference in the dietary changes between baseline and week 6, as well as between baseline and week 12, based on the intervention (X² test: X² = 2.86, *p* = 0.09 and X² = 0.70, *p* = 0.40, respectively (Table 4 and Table 5)).

## 4. Discussion

This study compared the effects of nutritional counselling on fasting and a whole-food plant-based diet to those of a DGE diet on the actual dietary behavior of patients with rheumatoid arthritis. Moreover, their respective effects on the nutrient supply were investigated.

Our cluster analysis across the dietary patterns did not reveal clusters that could clearly be associated with the two intervention groups. This does not necessarily indicate the inefficacy of the interventions or the dietary counselling. First, our study interventions improved RA disease activity to similar degrees in both groups [10]. Second, previous studies also support dietary interventions’ effectiveness and their impact on eating behaviors. This applies to both healthy participants as well as patients with morbidities [24]. Our results thus rather emphasize the similar effects of a whole-food PBD and a DGE diet on patients with RA based on this exploratory data set.

The intricate influences of nutrition on health and disease are challenging to study. For a better understanding, they may be broken down into the direct and indirect cause–effect relationships of dietary components: turmeric, for instance, directly inhibits the production of proinflammatory cytokines such as TNF-α in several in vitro and in vivo studies [19,25]. In contrast, complex nondigestible carbohydrates indirectly affect the immune system after being fermented to short-chain fatty acids (SCFA) by the gut microbiota. SCFA dampen NF-κB signaling, and thereby the secretion of proinflammatory cytokines, and induce the differentiation of regulatory T-cells in the colonic epithelium. As such, they contribute to a local tolerogenic immune environment [26,27,28]. The indirect mechanisms of diet are of particular interest given the current debate on the contribution of the gut microbiota to the pathogenesis of immune-mediated diseases. With respect to RA in particular, the hypothesis of a gut–joint axis is endorsed by a growing body of literature [29,30,31,32,33]. Wagenaar et al. recently investigated the effects of dietary interventions on chronic inflammatory diseases in relation to the gut microbiota [34]. Plant-based diets were reported to improve disease-specific outcomes in RA, which is in line with our results, but also type 2 diabetes and cardiovascular disease. Additionally, it was hypothesized that plant-based diets promote a more favorable composition of the microbiome [35,36]. However, the exact mechanisms are still unclear to date.

We did not study the further beneficial effects of fasting in this RCT, such as DNA repair, autophagy and enhanced immune function [37,38]. Fasting effects may also be partly mediated by an altered gut microbiota [39,40].

Regarding nutrient supply, both groups maintained their daily intake of carbohydrates, sugars, fiber and protein over the course of the study.

However, the fasting and PBD group indicated a reduced energy intake by approximately 100 kcal during the study period compared with the baseline. Although this trend was not statistically significant, it may still have produced a clinically relevant effect over 3 months. One possible explanation for the weight loss of approximately 5% in the fasting and PBD groups is the relative caloric deficit [10]. However, the intermittent fasting regimen in the PBD group itself may also have accounted for the observed weight loss [41,42].

With respect to macronutrients, the fasting and PBD group consumed significantly less total saturated fat than the DGE group, which favors a protective cardiovascular profile [43]. Interestingly, Sebe et al. have associated the consumption of saturated fat with muscle loss in a prospective cohort study with RA patients, as well as in a murine model [44].

Though not significant, we observed a trend towards higher intake of unsaturated fatty acids in the fasting and PBD group. An enhanced intake of such fats has been associated with a beneficial response in terms of RA disease activity [45].

The average daily intake of protein in the DGE group exceeded that of the fasting and PBD group and even the daily recommended intake. Epidemiological studies underscore that a high intake of animal protein, particularly red meat, may be related to the promotion of age-related diseases and mortality. This association seems to be abolished or attenuated when proteins are plant-derived [46,47].

As for micronutrients, the DGE group had a greater intake of vitamin A, vitamin B12, vitamin D, riboflavin and calcium (each *p* ≤ 0.02). Zinc and calcium were below the recommended daily allowances in both groups.

This study has several strengths. The parallel study design minimized unanticipated variation during the study period. With the COVID-19 pandemic, we changed personal coaching to remote consultations. Both groups were equally exposed to the matched intervention modality. In addition, the presented data are derived from a prospective intervention rather than an observational study, which renders them more informative.

The present study also has several limitations. First, the food records were self-recorded by the study participants, which required intensive participant involvement. As a result, the quality of these data might vary interindividually and represents a source of information bias. Specifically, dietary intake was adjusted for the physical activity level of the participant, but no other method was applied for the plausibility verification of this self-reported item. The parallel study design is a strength; on the other hand, this study design requires a larger sample size than, for example, a crossover design to achieve similar statistical power [48]. Hence, the moderate sample size in our exploratory trial might have been too small to demonstrate clear clustering. Furthermore, the number of patients included in the analysis here is relatively small, within the framework of an already rather small exploratory trial. This issue is attributable to (a) the COVID-19 pandemic and a slower recruitment rate than expected and (b) partly incomplete dietary data sets. Moreover, the analysis was unintentionally performed with data from female study participants only, leading to a heavy gender bias in this study. This imbalance is typical for studies of rheumatoid arthritis and studies on dietary interventions and fasting, as in both cases most patients are female. Therefore, our findings might not apply to male patients with RA in the same way.

## 5. Conclusions

The results of this study suggest that dietary counselling for a whole-food PBD including TRE compared to a guideline-based anti-inflammatory diet does not lead to two different dietary clusters, meaning two distinct dietary behaviors, as expected. This might explain why both groups performed similarly regarding the observed disease activity.

However, in terms of nutrient supply a whole-food plant-based diet including TRE might result in micronutrient deficiencies in vitamin A, vitamin B12, vitamin D, riboflavin and calcium compared to a guideline-based anti-inflammatory diet for RA. Micronutrient deficiencies can and should therefore be avoided through individualized, regular nutritional counselling and education.

Further confirmatory studies with larger sample sizes are needed to further define dietary recommendations for patients with rheumatoid arthritis.

## Figures and Tables

**Figure 1 nutrients-15-00851-f001:**
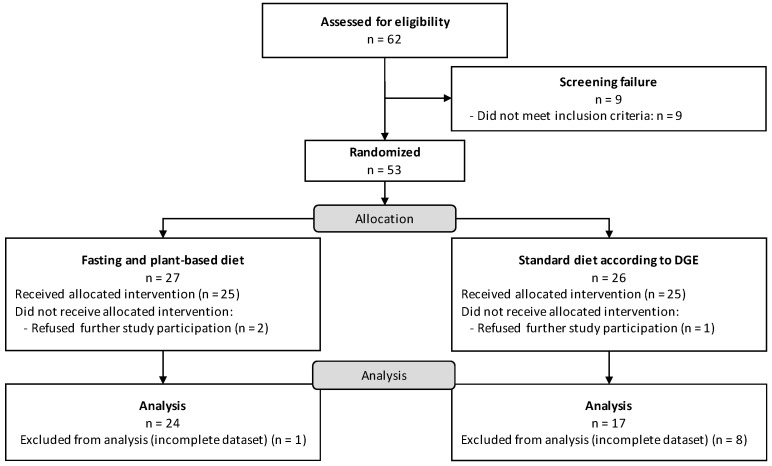
CONSORT flow diagram.

**Figure 2 nutrients-15-00851-f002:**
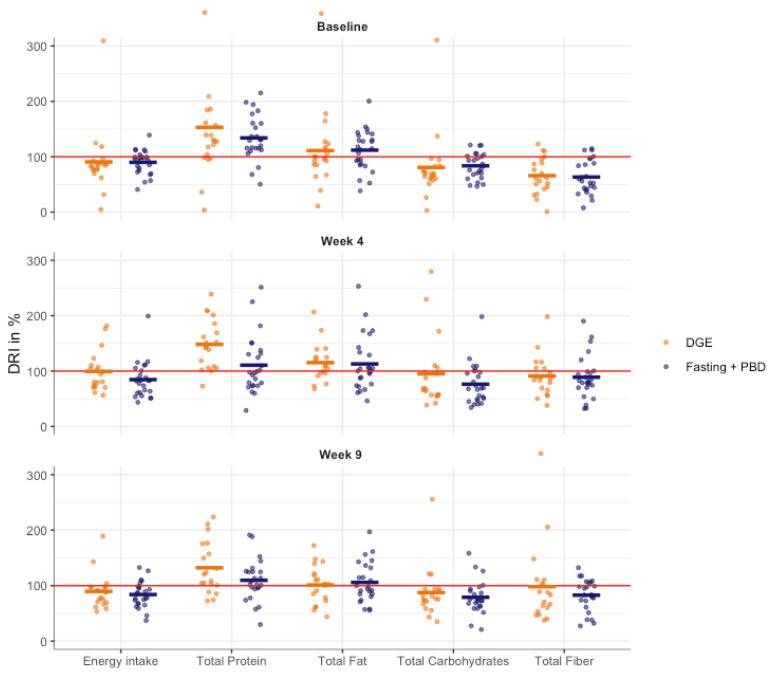
Dietary intake displayed as percentage of adequate nutrient intake according to the D-A-CH reference values at study weeks -1, 4 and 9: energy intake, total protein, fat, carbohydrate and fiber intake. DGE, German Nutrition Society; DRI, dietary recommended intake; PBD, plant-based diet.

**Table 1 nutrients-15-00851-t001:** Nutrient supply of participants with RA at study week -1 as well as the DGE reference values for adult women [22].

Nutrients	Overall Mean ± SD (*n* = 41)	Fasting + PBD (*n* = 24)	DGE (*n* = 17)	Reference Values	*p*-Value
Energy intake, kcal/day	1932.1 ± 879.7	1880.5 ± 466.4	2005.0 ± 1269.8	2100.0	0.49
Carbohydrates, g/day	215.7 ± 112.8	214.8 ± 60.7	216.8 ± 162.8	256.1	0.21
Sugar, g/day	80.0 ± 42.0	81.9 ± 43.1	77.4 ± 42.0	61.0	0.78
Total fat, g/day	78.9 ± 37.1	76.7 ± 25.7	82.2 ± 49.7	69.2	0.73
Saturated	26.4 ± 11.4	24.8 ± 9.6	28.7 ± 13.6	16.2–23.1	0.46
Monounsaturated	19.11 ± 9.5	19.5 ± 8.9	18.6 ± 10.8	34.6–46.2	0.58
Polyunsaturated	13.6 ± 8.5	14.6 ± 9.2	12.2 ± 7.4	18.5–27.7	0.60
Protein, g/day	69.0 ± 35.8	63.0 ± 18.4	77.5 ± 50.9	48.0	0.51
Dietary fiber, g/day	22.6 ± 11.0	21.9 ± 10.8	23.7 ± 11.6	35.1	0.54
Vitamin E, mg/day	11.5 ± 9.3	11.3 ± 11.2	11.7 ± 6.0	14.0	0.24
Vitamin C, mg/day	118.1 ± 86.2	113.6 ± 67.0	124.5 ± 109.8	95.0	1.00
Vitamin D, µg/day	3.3 ± 4.2	2.2 ± 2.0	4.8 ± 5.8	2.0–4.0	0.25
Thiamine, mg/day	0.8 ± 0.6	0.8 ± 0.6	0.9 ± 0.6	1.0	0.65
Riboflavin, mg/day	0.9 ± 0.5	0.8 ± 0.4	1.1 ± 0.5	1.1	**0.02**
Vitamin B_6_, mg/day	1.1 ± 0.6	1.0 ± 0.4	1.3 ± 0.7	1.4	0.09
Biotin, µg/day	25.2 ± 15.6	25.6 ± 17.9	24.7 ± 12.5	40.0	0.52
Folate, µg/day	190.8 ± 97.1	176.7 ± 81.2	210.2 ± 116.2	300.0	0.40
Vitamin B_12_, µg/day	3.1 ± 3.9	1.8 ± 1.4	5.0 ± 5.4	4.0	0.05
Vitamin A, mg/day	0.3 ± 0.2	0.2 ± 0.2	0.3 ± 0.2	0.7	0.87
Vitamin K, µg/day	57.4 ± 46.0	59.5 ± 53.9	54.5 ±33.9	60.0	0.90
Zinc, mg/day	21.6 ± 85.8	31.8 ± 112.7	7.4 ± 6.2	8.0	0.60
Magnesium, mg/day	696.5 ± 2306.9	982.6 ± 3028.3	300.3 ± 141.4	300.0	0.46
Copper, µg/day	3422.0 ± 11557.9	4823.3 ± 15178.5	1481.8 ± 747.8	1000–1500	0.40
Calcium, mg/day	1625.9 ± 5734.2	2306.1 ± 7536.5	684.1 ± 291.1	1000.0	0.16

Values reported are means ± SD or *n* (%) unless specified otherwise. Reference values are displayed for orientation only. Significant values (*p* ≤ 0.05) of inter-group comparisons are bolded. DGE, German Nutrition Society; kcal, kilocalorie; PBD, plant-based diet.

**Table 4 nutrients-15-00851-t004:** Cluster analysis on baseline versus week 6.

	Cluster 1	Cluster 2	Total
DGE	17	6	23
Fasting + PBD	15	9	24
Total	32	15	47

**Table 5 nutrients-15-00851-t005:** Cluster analysis on baseline versus week 12.

	Cluster 1	Cluster 2	Total
DGE	22	1	23
Fasting + PBD	19	5	24
Total	41	6	47

## Data Availability

Data are available upon request. Requests for data sharing can be made by contacting the corresponding author. Data will be shared after review and approval by the trial scientific board, and terms of collaboration will be reached together with a signed data access agreement.

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
