# Peer review of "Post Hoc Analysis of a Randomized Controlled Trial on Fasting and Plant-Based Diet in Rheumatoid Arthritis (NutriFast): Nutritional Supply and Impact on Dietary Behavior"

_nutrients, 2023, doi:10.3390/nu15040851_

Round 1

Reviewer 1 Report

The results of a randomised clinical trial on the effect of two different diets on rheumatoid arthritis have already been published by the authors (Hartmann AM et al.: To eat or not to eat- an exploratory randomized controlled trial on fasting and plant-based diet in rheumatoid arthritis (NutriFast-Study). Frontiers in Nutrition 2022). After twelve weeks of diet, there was no difference in clinical response between the two groups.

The post-hoc analysis now presented compares the energy and nutrient intake of the study participants. Using cluster analysis, the two diets cannot be distinguished from each other despite differences in the intake of individual nutrient components.

The paper is interesting in my opinion, particularly because the primary assumptions of the authors seem to have been disappointed by the study results. The carefully recorded energy and nutrient intake of both diet groups at different time points provides a valuable base for possible further intervention studies.

Minor specific comments:

1.       "German Nutrition Society (DGE)" only needs to be explained once (lines 24, 59, 97/98).

2.       “physical activity level of 1.6 (lines 125/126)” - A brief explanation of the value for readers who are not familiar with the term would be helpful.

3.       Lines 133-136: As I have understood the protocol, the changes in diet between baseline and weeks 4 and 9 were recorded.

4.       Lines 141,154,158,159, 194: Here, at least in the text version I received, the references are not displayed.

5.       “…both groups maintained their daily intake of energy,…” – “In terms of energy intake, the fasting and PBD group reduced energy intake by approximately…” (lines 231-234): There is a contradiction here that should be resolved.

Reviewer 2 Report

The aim of this manuscript is to compare nutrient intake and nutritional behaviour of different diets in patients with rheumatoid arthritis.  

My comments are follows:

Abstract:

L28- unit of measurement is missing; L29-  vitamin B12- please use subscript- in Table 1. do the same;  L29- please correct the notation of the p-value,

Introduction:

L51: the author mentioned  that unsaturated fatty acids seem to lead to clinical improvements- Please rethink it, because n-3 fatty acids have beneficial effects, but n-6 fatty acids may play a role in inflammation.

Methods:

L96:  Please mention some spices and herbs that are known for their anti-inflammatory properties.

It's not clear how you assessed the plausibility of the energy intake?

Results

L141 and L154, L158, L159: Please check the references.

Table 1: The name of the second column is not correct, it shows the average of the 2 groups 

L236-237- the authors reported weight loss of about 5%, but the result did not include this.

 Please reconsider the explanation for the weight loss, there was no remarkable difference in energy intake between the groups.

 Lines 249-251 needs to be removed. It is not clear why low protein intake is described when it reached the reference value in all groups in all study periods.

 In the discussion/conclusion, the authors pointed to the results of another study, which also examined the activity of the disease- the authors should rather focus on the results of this study. 

Improve the conclusion - Dietary counselling can prevent micronutrient deficiencies.

Round 2

Reviewer 2 Report

Dear authors, your corrections have raised the scientific quality of the article! Congratulations! 

Author Response

Dear reviewer and collegue,

many thanks for the appreciation and your contribution to our article!